# Carbohydrate Assimilation and Translocation Regulate Grain Yield Formation in Wheat Crops (*Triticum aestivum* L.) under Post-Flowering Waterlogging

**Shangyu Ma [1,2,3], Panpan Gai [1], Yanyan Wang [4], Najeeb Ullah [5,6], Wenjing Zhang [1,2], Yonghui Fan [1,2], Yajing Shan [1], Zhenglai Huang [1,2,3] and Xia Hu [1,\***

[1] Department of Agronomy, Anhui Agricultural University, Hefei 230036, China; mashangyu@ahau.edu.cn (S.M.); 103-5664410@ahau.edu.cn (P.G.); zhangwenjing79@ahau.edu.cn (W.Z.); yonghuif@ahau.edu.cn (Y.F.); ydan4447@gmail.com (Y.S.); huangzhenglai@ahau.edu.cn (Z.H.)
[2] Key Laboratory of Wheat Biology and Genetic Improvement on Southern Yellow & Huai River Valley, Ministry of Agriculture and Rural Affairs, Hefei 230036, China
[3] Jiangsu Collaborative Innovation Center for Modern Crop Production, Nanjing 210095, China
[4] Department of Agricultural Park Management Center, Anhui Agricultural University, Hefei 230036, China; wangyanyan@ahau.edu.cn
[5] Queensland Alliance for Agriculture and Food Innovation (QAAFI), The University of Queensland, Toowoomba, QLD 4350, Australia; n.ullah@uq.edu.au
[6] Faculty of Science, Universiti Brunei Darussalam, Jalan Tungku Link, Gadong BE1410, Brunei
**\*** Correspondence: huxia2011@ahau.edu.cn

**Abstract:** In a two-year field study, we quantified the impact of post-flowering soil waterlogging on carbon assimilation and grain yield formation in wheat crops. At anthesis, wheat cultivars YangMai 18 (YM18) and YanNong 19 (YN19) were waterlogged for different durations, i.e., 0 (W0), 3 (W3), 6 (W6) and 9 (W9) days using artificial structures. Changes in leaf physiology, carbon assimilation and biomass production were quantified at 0, 7, 14, and 21 days after anthesis under all treatments. Short-term (W3) waterlogging had no significant effect on wheat crops but W6 and W9 significantly reduced the net photosynthetic rate (Pn), leaf SPAD value, and grain weight of the tested cultivars. Increasing waterlogging duration significantly increased dry matter accumulation in the spike-axis + glumes but reduced dry matter accumulation in grain. Further, the tested cultivars responded significantly variably to W6 and W9. Averaged across two years, YM 18 performed relatively superior to YN19 in response to long-term waterlogging. For example, at 14 days after anthesis, W9 plants of YM18 and YN19 experienced a 17.4% and 23.2% reduction in SPAD and 25.3% and 30.8% reduction in Pn, respectively, compared with their W0 plants. Consequently, YM18 suffered a relatively smaller grain yield loss (i.e., 16.0%) than YN19 (23.4%) under W9. Our study suggests that wheat cultivar YM18 could protect grain development from waterlogging injury by sustaining assimilates supplies to grain under waterlogged environments.

**Keywords:** wheat; waterlogging; carbon assimilation; photosynthetic characteristics; grain filling rate; yield

## 1. Introduction

Climatic variability typified by extreme weather events—particularly, erratic rainfall and concomitant soil waterlogging—poses serious challenges to crop production and global food security [1]. For instance, soil waterlogging affects 25% of the global wheatbelts [2], and in China alone, this damage accounts for 20–35% of the agricultural lands [3]. Wheat is one of the most important food crops in the world, but it suffers significant yield losses when cultivated on waterlogged soils [4]. Insufficient root oxygen ($O_2$) supply is a major waterlogging damage to crop plants [5–7], which inhibits root development [8], root-to-shoot ratio, and root functioning. Inhibited root growth and nutrient supplies affect growth

and functioning of above-ground plant parts [6]. In addition, root level hypoxia induces bioactive molecules such as nitric oxide [9] and reactive oxygen species such as hydrogen peroxide [10], which induce mitochondrial damage [11], leading to decreased amyloid numbers [12] and programmed cell death. The severity of stress and subsequent yield losses depend on several factors, such as stress duration [13], soil and climatic conditions [14], and the genetic background of the crops [11].

At the physiological level, changes in leaf net photosynthetic rate (Pn) are a sensitive indicator of waterlogging-induced damage in plants [15]. Under waterlogged environments, when root hydraulic conductivity is impaired, terrestrial plants tend to close their stomata, thereby affecting overall carbon and biomass assimilation [12–16]. Inhibited assimilate supplies, particularly during grain filling, can significantly reduce grain development and final grain weight [10]. Moreover, waterlogging significantly reduces the dry matter accumulation, remobilization, and yield; the extent of yield loss increased with the increase in waterlogging stress severity [17]. In terms of yield components, studies have shown that the numbers of ears per plant and grains per ear are the traits most affected by waterlogging, but relative sensitivity to waterlogging varies with the duration of waterlogging [18]. Research shows that long-term waterlogging, i.e., 44 days at 93 days after sowing and 58 days at 64 days after sowing, decreased grain yield by 20% and 24%, respectively [19].

Significant genotypic variations in soil waterlogging tolerance have been reported in different crops, including wheat. For example, prolonged soil waterlogging during three and four leaf growth stages (Zadoks stages 13 and 14) had no significant effect on a wheat cultivar Blasco, but it reduced grain yield of cultivar Aquilante by 27% [20]. A reduced number of spikes per plant and spikelets per spike were attributed to this grain yield loss under sustained waterlogging. Similarly, Bao [21] suggested the following order of waterlogging sensitivity in wheat during different developmental stages: booting stage > jointing stage > tillering stage > grain filling stage. Furthermore, waterlogging during different developmental stages, variably affected the grain yield components of wheat, i.e., grain number during the stem elongation and booting stages and 1000-kernel weight during the grain filling stage [22]. The anthesis and grain filling phases in wheat crop are relatively more susceptible to soil waterlogging [23] than its vegetative phases [24].

Previous studies mainly focus on the impact of waterlogging on different growth periods of wheat, and the effect of different waterlogging durations during anthesis is rarely reported. This study quantifies how different soil waterlogging durations affect the dynamic of grain growth in wheat crops. The study also explores the link between post-flowering assimilates supplies and waterlogging tolerance in wheat genotypes of contrasting sensitivity.

## 2. Materials and Methods

### 2.1. Experiment Site

Field experiments were carried out at the experiment station of Anhui Agricultural University (117.01′ E, 30.57′ N) in Lujiang County, Hefei City, Anhui Province during 2015–2016 and 2016–2017 wheat-growing seasons. This location has a humid monsoon climate in the northern subtropics. The average annual precipitation is 1324.89 mm. The average precipitation in the past 10 years and the monthly precipitation in the growing seasons are shown in Figure 1. The soil nutrient contents at depth of 0–20 cm before planting is shown in Table 1.

### 2.2. Experiment Design

Wheat cultivars YangMai 18 (YM18, moderately waterlogging tolerant) and YanNong 19 (YN19, waterlogging sensitive) were used in these experiments. The seeds were sown at a density of 300 seeds per m$^2$ on 8 November 2015 and 11 November 2016. All plots were supplied with 225 kg N ha$^{-1}$, 75 kg P$_2$O$_5$ ha$^{-1}$, and 150 kg K$_2$O ha$^{-1}$. Total P and

K fertilizers and 70% of N fertilizers were applied before sowing, and the remaining N fertilizer was top-dressed at jointing.

The crop was planted in an experimental plot (2.4 m × 5 m) with a row spacing 20 cm, and each treatment was replicated three times in a randomized complete block design. The edge of the plot was insulated with a plastic frame made of polyvinyl chloride. The plastic frame was buried 40 cm deep and extended 20 cm on the ground. Waterlogging was carried out in each plot by an artificial method at flowering (Zadoks decimal growth stage [25], Z65). A 2 cm water layer above the ground was established for 0, 3, 6, and 9 days and mentioned as W0, W3, W6, and W9, respectively. At the end of each waterlogging period, the water was discharged from the plots and allowed to drain freely before irrigating again, which did not occur for 3–5 days following waterlogging treatment. W0 from sowing to maturity and the waterlogged plots before and after the treatments were irrigated as necessary to maintain 15–20% volumetric soil moisture (approximately 80% of field capacity).

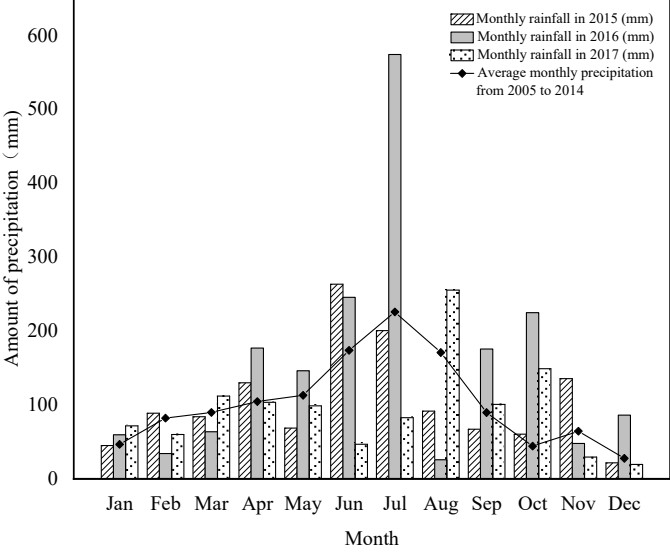

**Figure 1.** Monthly precipitation from 2015 to 2017 and average monthly precipitation from 2005 to 2014.

**Table 1.** Soil nutrient contents before sowing.

| Growing Season | Organic Matter (g·kg⁻¹) | Total N (g·kg⁻¹) | Available P (mg·kg⁻¹) | Available K (mg·kg⁻¹) | Available N (mg·kg⁻¹) |
|---|---|---|---|---|---|
| 2015–2016 | 18.5 | 1.2 | 12.1 | 127.4 | 133.5 |
| 2016–2017 | 17.3 | 1.2 | 11.7 | 142.9 | 118.5 |

*2.3. Measurements and Methods*

For each trial in the 2015–2016 and 2016–2017 growing season, approximately 100 stems of each waterlogging treatments were tagged at flowering (Z65). Greenness and gas exchange of flag leaf and spikes were measured from the tagged stem of each plot.

2.3.1. SPAD Value and Leaf Gas Exchange of Flag Leaf

All measurements were performed at 0, 7, 14, and 21 days after flowering from 09:30 to 11:30. Five flag leaves of the tagged stem were collected for each treatment each time [26]. Flag leaf greenness was measured nondestructively using a SPAD-502 Meter (Soil Plant Analysis Development, Minolta, Japan). Five SPAD readings were performed on each leaf and averaged [27].

The net photosynthetic rate (Pn) of flag leaves was measured using a portable photosynthesis system (LI-6400, LI-Cor, United States) at a $CO_2$ concentration of 385 mol, and light intensity of 1200 mol·m$^{-2}$·s$^{-1}$ [28].

### 2.3.2. Parameters of Wheat Grain Filling

The ten spikes tagged at anthesis were harvested from each plot starting from 7 days after anthesis until maturity with a 7-day interval. The spikes were oven-dried at 70 °C to constant weight, hand threshed and grain counted and weighed [29]. The grain- filling rate was estimated from the accumulation of dry grain weight.

The number of days after anthesis ($t$) was treated as an independent variable, and 1000-grain weight ($Y$) was measured each time as a dependent variable. The logistic equation $Y = K/(1 + e^{(A+Bt)})$ was used to determine the grain growth process. $K$ is the fitted maximum grain weight that can be achieved at the end of grain filling stage, $A$ is related to both the duration and rate of grain filling, $B$ is related to the rate of the grain filling, and $A$ and $B$ are constants. The coefficient of determination ($R^2$), the ratio of the regression sum of squares of $Y$ in accordance with $t$ to the total sum of squares, is used to indicate its goodness of fit. According to the logistic equation, first and second derivatives of the equation, a series of filling parameters was derived [30].

(1)  Start date of the peak grain filling period: $t_1 = [A - ln(2 + 1.732)]/(-B)$.
(2)  End date of the peak grain filling period: $t_2 = [A + ln(2 + 1.732)]/(-B)$.
(3)  End of grain filling ($Y$ up to 99% K) date: $t_3 = (4.595 - 12 + A)/(-B)$.
(4)  Date when the maximum grain filling rate appears: $Tm = -A/B$, and the maximum grain filling rate $Vm = -BK/4$.
(5)  $T_1$, $T_2$, and $T_3$ represent the grain filling rates of the gradual, rapid, and slow increase stages, respectively. $T_1 = t_1$, $T_2 = t_2 - t_1$, and $T_3 = t_3 - t_2$.
(6)  Grain filling duration: $T = t_3$, and mean grain filling rate: $Va = K/t$.

### 2.3.3. Dry Matter Accumulation and Distribution

At the crop maturity, 20 plants were harvested manually at the ground level from each plot. These plants were separated into leaves, stems + sheaths, spike axis + glumes, and grains. All samples were dried to a constant weight in a forced-draft oven at 70 °C to get dry weights.

### 2.3.4. Grain Yield and Components

The spike numbers of three plots were investigated. Thirty spikes were threshed to determine the kernels per spike and 1000-grain weight. Grain yield was determined from each plot by harvesting 2 m × 1 m quadrant cuts per sampling plot at 13% moisture grain content.

### 2.4. Data Analysis

The software Data Processing System 16.0 (DPS, China) was used for all statistical analyses. An analysis of variance (ANOVA) was used to determine the significance of cultivar, waterlogging, and growing season of the study, on SPAD, net photosynthetic rate (Pn), and 1000-grain weight (GW) according to the model of a three-way analysis. For the SPAD and Pn of flag leaves, dry matter accumulation and distribution, grain yield, and grain components, the significance of cultivar and waterlogging was determined through two-way ANOVA. Multiple comparisons were made using the least significant difference test with $\alpha = 0.05$ to determine significant effects among treatments. The Origin9.1 (Origin-Lab, Northampton, MA, USA) was used to show differences in SPAD values, Pn values, grain weight, and grain filling rate.

## 3. Results

### 3.1. SPAD of Flag Leaves

As the duration of waterlogging increased, it reduced flag leaf SPAD values in both wheat cultivars. An analysis of the data pooled across two cultivars showed that waterlogging significantly affected all the growth parameters of wheat at 7, 14 and 21 days after flowering (Table 1). There was no significant effect of W3 on leaf SPAD values of tested wheat cultivars during both years. However, SPAD values were significantly reduced

in response to W6 and W9 treatments throughout the grain filling period (i.e., 7, 14 and 21 days after flowering), although the difference between W6 and W9 was not significant at 7 days after flowering.

Averaged across two years, W6 and W9 reduced leaf SPAD by 10.7% and 11.7%, respectively in YM18 and by 12.5% and 12.7%, respectively, in YN19 compared with their respective W0 plants at 7 days after flowering (Figure 2). The genotypic variation in response to waterlogging became significant when the SPAD value was measured at 14 days after flowering. Compared with W0, W6 and W9 reduced leaf SPAD of YM18 by 8.8% and 17.4% (averaged across two years), respectively. In contrast, this reduction was greater for YN19, where W6 and W9 caused a 20.8% and 24.3% (averaged across two years) reduction in leaf SPAD, respectively. The gap between leaf SPAD value of waterlogged (W6 and W9) and control further grew wider when tested at 21 days after flowering; i.e., W9 reduced leaf SPAD by 34.3% and 33.2% (averaged across two years) in YN19 and YM18, respectively.

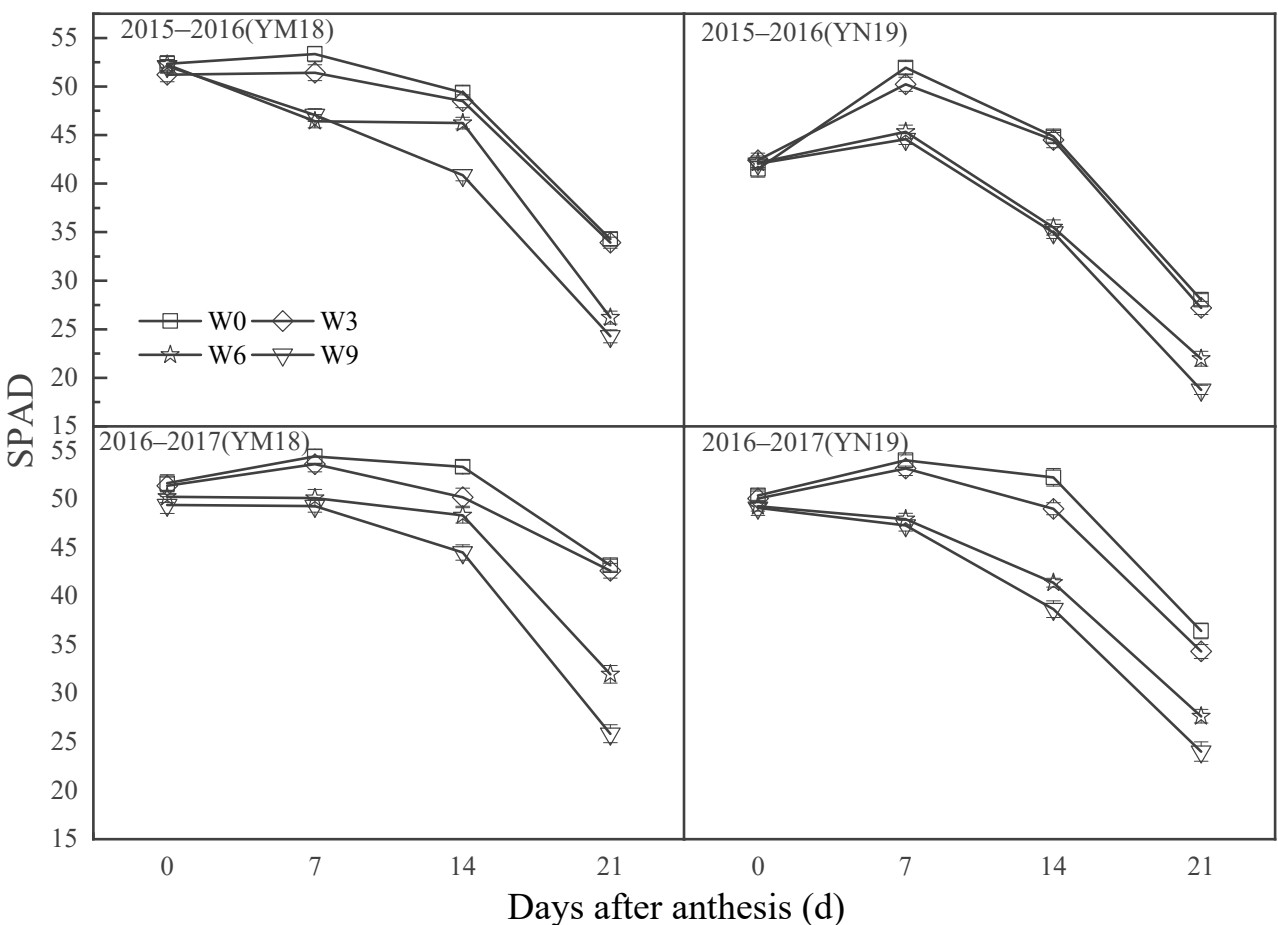

**Figure 2.** SPAD values of flag leaves under different treatments. Wheat genotypes YangMai18 (YM18) and YanNong19 (YN19) were subjected to soil waterlogging at anthesis for different durations and data were collected at 0, 7, 14 and 21 days after anthesis. W0 = control, W3 = 3 days of waterlogging, W6 = 6 days of waterlogging, W9 = 9 days of waterlogging. Each data point represents the mean ± SE of three independent replicates.

Significant cultivar × waterlogging for leaf SPAD values were observed both at 14 and 21 days after anthesis (Table 2), with YM18 performing relatively superior to YN19 under waterlogging (Figure 2). Year×waterlogging interactions were significant for leaf SPAD values when measured at 21 days after anthesis (Table 2). Wheat cultivars particularly under W3 had a relatively more leaf SPAD at this stage during 2016–2017 than 2015–2016 (Figure 2).

**Table 2.** ANOVA (*p*-value) for the effects of cultivars, waterlogging, and growing seasons and their interactions on some shoot and yield traits.

| Treatments | 7 Days after Flowering | | | 14 Days after Flowering | | | 21 Days after Flowering | | |
|---|---|---|---|---|---|---|---|---|---|
| | SPAD | Pn | GW | SPAD | Pn | GW | SPAD | Pn | GW |
| Years | **<0.001** | 0.240 | **0.002** | **<0.001** | **<0.001** | 0.068 | **<0.001** | **0.023** | **0.001** |
| Cultivar | **0.001** | **0.043** | 0.708 | **<0.001** | **0.008** | **0.001** | **<0.001** | **<0.001** | **<0.001** |
| Waterlogging | **<0.001** | **<0.001** | **<0.001** | **<0.001** | **<0.001** | **<0.001** | **<0.001** | **<0.001** | **<0.001** |
| Years × Cultivar | 0.691 | 0.095 | 0.248 | **0.001** | **0.001** | 0.099 | 0.624 | **<0.001** | 0.307 |
| Years × Waterlogging | 0.434 | 0.938 | 0.489 | 0.092 | 0.540 | 0.839 | **<0.001** | 0.062 | **0.005** |
| Cultivar × Waterlogging | 0.412 | 0.342 | 0.411 | **<0.001** | 0.974 | 0.967 | **0.002** | 0.939 | 0.235 |
| Years × Cultivar × Waterlogging | 0.777 | 0.295 | 0.501 | 0.267 | 0.831 | 0.613 | 0.081 | 0.568 | 0.302 |

Data presented in the table were collected at 7 days, 14 days and 21 days after flowering and summarized the significant differences (*p* values). The significant (*p* < 0.05) effects are shown as bold in the ANOVA table. Pn = Net photosynthetic rate ($\mu$mol $CO_2$ m$^{-2}$ s$^{-1}$); GW = 1000-grain weight (g).

### 3.2. Net Photosynthetic Rate of Flag Leaves

No significant effect of short-term waterlogging (W3) was observed on the net photosynthetic rate (Pn) of flag leaves of two cultivars at 7 and 14 days after flowering (Figure 3). However, W6 and W9 significantly reduced the Pn of the two cultivars during all the tested developmental stages. Averaged across all treatments, wheat cultivars had a maximum Pn at 7 days after anthesis, which was reduced during later measurements (14 and 21 days after anthesis).

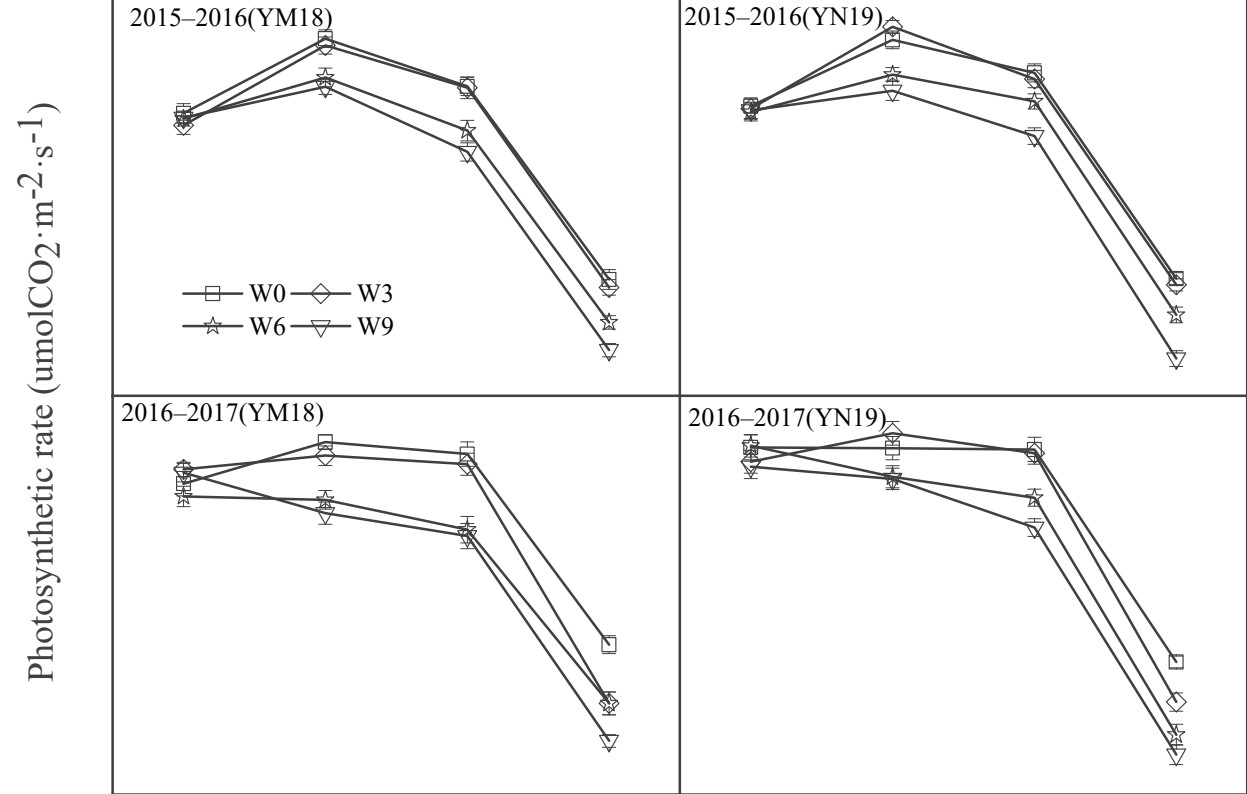

Days after anthesis (d)

**Figure 3.** Net photosynthetic rates of flag leaves of wheat under different treatments. Wheat genotypes YangMai18 (YM18) and YanNong19 (YN19) were subjected to soil waterlogging at anthesis for different durations and data were collected at 0, 7, 14 and 21 days after anthesis. W0 = control, W3 = 3 days of waterlogging, W6 = 6 days of waterlogging, W9 = 9 days of waterlogging. Each data point represents the mean ± SE of three independent replicates.

Averaged across two years, W6 and W9 reduced Pn by 8.2% and 18.1% in YM18 and by 11.0% and 19.1% in YN19, respectively, compared with control (Figure 3). When measured at 21 days after flowering, reductions of Pn of YM18 under W6 and W9 were 24.3% and 38.4%, respectively, during 2015–2016, and 26.3% and 41.8%, respectively, during 2016–2017. Meanwhile the Pn of YN19 under W6 and W9 was reduced by 22.6% and 48.4%, respectively, during 2015–2016, and 39.0% and 53.1%, respectively, during 2016–2017. On average, YM18 had a relatively higher net photosynthetic rate of flag leaves under all the treatment than that of YN19 (Table 2).

### 3.3. Grain Weight

On average, YM18 produced significantly larger grains (measured in terms of 1000-grain weight, TGW) than YN19 across different treatments and developmental stages except at 7 days after anthesis during 2015–2016 (Table 2, Figure 4). W3 had no significant effect on TGW but W6 and W9 significantly reduced TGW at 14, 21 and 28 days after anthesis in YM18. TGW of under W6 and W9 treatments was significantly lower than that of the W3 and W0 treatments at 14, 21, and 28 days after anthesis, and no significant differences were observed in TGW under W6 and W9 treatments at 14 and 28 days after anthesis in YN19 (Figure 4).

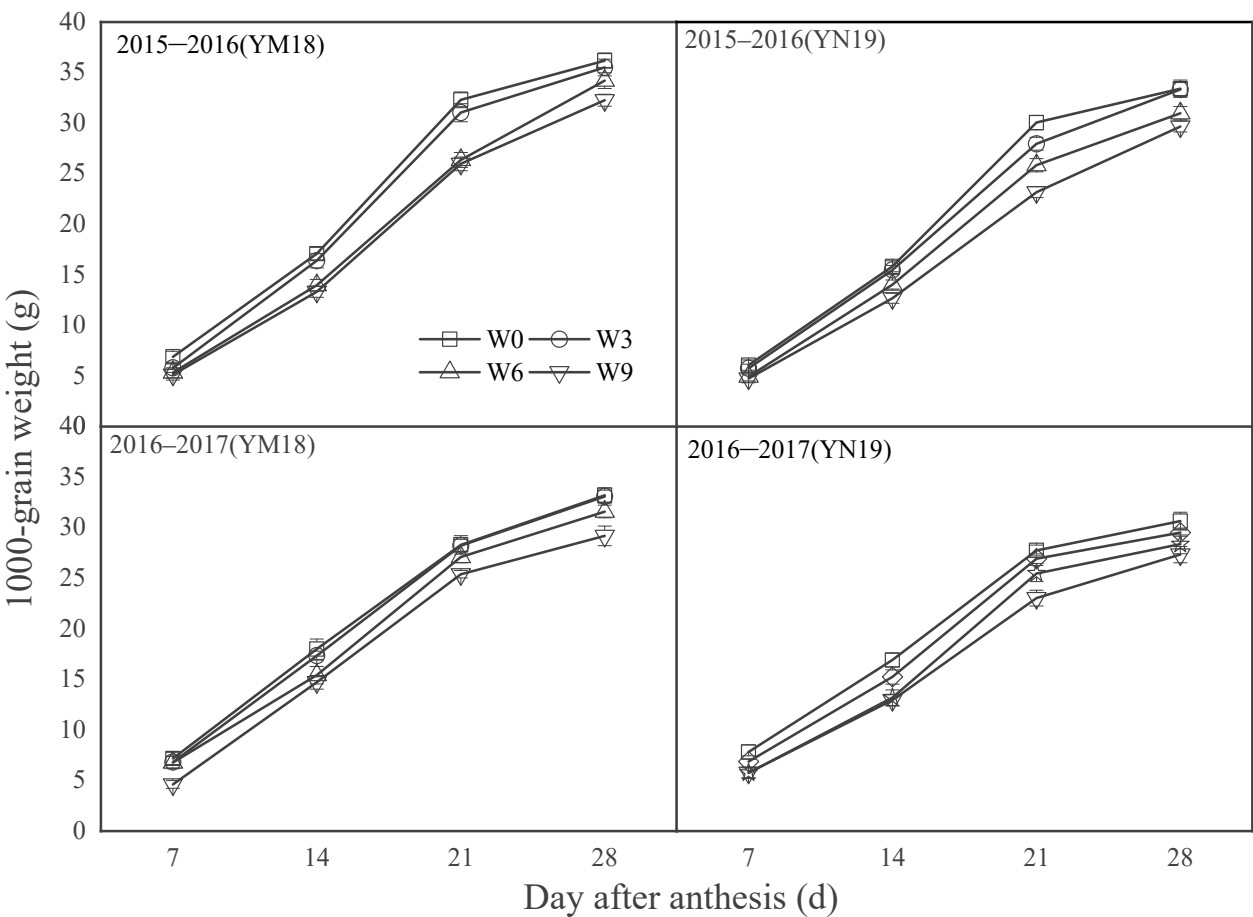

**Figure 4.** Grain weight of wheat under different treatments. Wheat genotypes YangMai18 (YM18) and YanNong19 (YN19) were subjected to soil waterlogging at anthesis for different durations and data were collected at 0, 7, 14 and 21 days after anthesis. W0 = control, W3 = 3 days of waterlogging, W6 = 6 days of waterlogging, W9 = 9 days of waterlogging. Each data point represents the mean ± SE of three independent replicates.

### 3.4. Grain Filling Characteristic Parameters

Coefficients of determination of grain weight for each equation were highly significant in both growing seasons, suggesting that the logistic equation could accurately describe the grain filling process of wheat cultivars in this experiment (Table 3). Analysis of the grain filling characteristic parameters of different treatments showed that the TGW, maximum grain filling rate (Tm), grain filling duration (T), rapid increase stage ($T_2$), and slow increase stage ($T_3$) in YM18 were all positively correlated with waterlogging duration and negatively correlated with the maximum grain filling rate (Vm), mean grain filling rate (Va), and gradual increase stage ($T_1$). In YN19, Vm, and Va were positively correlated with waterlogging duration and negatively correlated with Tm, T, T1, T2, and T3. Furthermore, compared with W0, under W9, Vm, Va and T1 of YM18 were increased by 40.7%, 13.5%, and 28.6%, respectively, while T and T3 of YM18 were decreased by 24.1% and 36.3%, respectively. However, under W9, Vm and Va of YN19 were decreased by 18.4% and 18.6%, respectively while T, T1, and T2 of YN19 were increased by 8.6%, 12.1%, and 10.8%, respectively (Table 3).

**Table 3.** Logistic equation and characteristic parameters of grain filling in different treatments.

| Growing Season | Cultivar | Treatment | Logistic Equation | $R^2$ | Tm (d) | Vm (mg·grain$^{-1}$·d$^{-1}$) | T (d) | Va (mg·grain$^{-1}$·d$^{-1}$) | T1 (d) | T2 (d) | T3 (d) |
|---|---|---|---|---|---|---|---|---|---|---|---|
| 2015–2016 | YM18 | W0 | $Y = 38.6399/(1 + e^{2.7097-0.172703t})$ | 0.9916 | 15.7 | 1.7 | 42.3 | 0.9 | 8.1 | 14.7 | 19.0 |
| | | W3 | $Y = 37.9719/(1 + e^{2.9108-0.182388t})$ | 0.9936 | 16.0 | 1.7 | 41.2 | 0.9 | 8.7 | 15.0 | 18.0 |
| | | W6 | $Y = 35.2239/(1 + e^{4.1522-0.278026t})$ | 0.9897 | 14.9 | 2.5 | 31.5 | 1.1 | 10.2 | 13.9 | 11.8 |
| | | W9 | $Y = 33.1104/(1 + e^{5.1911-0.355704t})$ | 0.9728 | 14.6 | 2.9 | 27.5 | 1.2 | 10.9 | 13.6 | 9.2 |
| | YN19 | W0 | $Y = 34.7171/(1 + e^{3.4524-0.242375t})$ | 0.9950 | 14.2 | 2.1 | 33.2 | 1.1 | 8.8 | 13.3 | 13.5 |
| | | W3 | $Y = 34.6754/(1 + e^{3.2611-0.221364t})$ | 0.9986 | 14.7 | 1.9 | 35.5 | 1.0 | 8.8 | 13.7 | 14.8 |
| | | W6 | $Y = 32.1739/(1 + e^{3.4408-0.229923t})$ | 0.9993 | 15.0 | 1.9 | 35.0 | 0.9 | 9.2 | 14.0 | 14.3 |
| | | W9 | $Y = 31.0240/(1 + e^{3.2146-0.206115t})$ | 0.9969 | 15.6 | 1.6 | 37.9 | 0.8 | 9.2 | 14.6 | 15.9 |
| 2016–2017 | YM18 | W0 | $Y = 36.2162/(1 + e^{2.6678-0.187727t})$ | 0.9991 | 14.2 | 1.7 | 38.7 | 0.9 | 7.2 | 13.2 | 17.5 |
| | | W3 | $Y = 35.8357/(1 + e^{2.7933-0.194550t})$ | 0.9996 | 14.4 | 1.7 | 38.0 | 0.9 | 7.6 | 13.4 | 16.9 |
| | | W6 | $Y = 34.1698/(1 + e^{2.8480-0.194831t})$ | 0.9980 | 14.6 | 1.7 | 38.2 | 0.9 | 7.9 | 13.6 | 16.8 |
| | | W9 | $Y = 30.6619/(1 + e^{3.3618-0.234306t})$ | 0.9998 | 14.4 | 1.8 | 34.0 | 0.9 | 8.7 | 13.4 | 14.0 |
| | YN19 | W0 | $Y = 33.3013/(1 + e^{2.5785-0.191689t})$ | 0.9962 | 13.5 | 1.6 | 37.4 | 0.9 | 6.6 | 12.5 | 17.1 |
| | | W3 | $Y = 32.7110/(1 + e^{2.7763-0.195348t})$ | 0.9925 | 14.2 | 1.6 | 37.7 | 0.9 | 7.5 | 13.2 | 16.8 |
| | | W6 | $Y = 30.6996/(1 + e^{3.1501-0.214451t})$ | 0.9927 | 14.7 | 1.7 | 36.1 | 0.9 | 8.6 | 13.7 | 15.3 |
| | | W9 | $Y = 29.5724/(1 + e^{2.8629-0.192369t})$ | 0.9981 | 14.9 | 1.4 | 38.8 | 0.8 | 8.0 | 13.9 | 17.0 |

Note: Y, 1000-grain weight; $R^2$, determination coefficient; Tm, time reaching the maximum grain-filling rate; Vm, maximum grain filling rate; T, duration of grain filling stage; Va, mean grain filling rate; T1, duration of the gradual increase in grain filling rate stage; T2, duration of the rapid increase in grain filling rate stage; and T3, duration of the slow increase grain filling rate stage.

These results showed that the grain weight decreased with the increasing waterlogging duration, but the reduction in these parameters was cultivar specific in this study.

### 3.5. Dry Matter Accumulation and Distribution

Short-term waterlogging (W3) had no significant effect on dry matter distribution in the tested wheat cultivars during both growing seasons. However, long-term waterlogging (W6 and W9) significantly reduced dry matter accumulation in grains and stem + sheath tissue but increased in spike axis + glume (Table 4).

Compared with W0 plants, a maximum reduction in dry matter accumulation in grains was observed in YN19, under W9, i.e., 27.9% and 20.0% during 2015–2016 and 2016–2017, respectively. In YM18, W9 caused a maximum reduction in dry matter accumulation in stem + sheaths (15.2%, 2015–2016) and the proportion of dry matter in stem + sheaths to total dry matter accumulation (9.2%, 2015–2016), and a maximum increase in dry matter accumulation in spike axis + glumes. Averaged across waterlogging treatments, YM18 exhibited 17.0%, 16.0%, 21.1% and 28.6% higher distribution of dry matter to grains compared with YN19 under W0, W3, W6, and W9, respectively. Meanwhile, the proportion of dry matter accumulation in grains to total dry matter accumulation of YM18 was 10.6%, 13.2%, and 17.0% higher than those of YN19 under W3, W6, and W9, respectively (Table 4). These results suggested a higher efficiency of YM18 in remobilizing assimilates towards developing grains than YN19 under waterlogged environments.

**Table 4.** Dry matter accumulation and its distribution at maturity stage under different treatments.

| Growing Season | Cultivar | Treatment | Grain | | Stem + Sheath | | Spike Axis + Glume | |
|---|---|---|---|---|---|---|---|---|
| | | | Amount (kg·hm$^{-2}$) | Proportion of the Total (%) | Amount (kg·hm$^{-2}$) | Proportion of the Total (%) | Amount (kg·hm$^{-2}$) | Proportion of the Total (%) |
| 2015–2016 | YM18 | W0 | 7380.8 a | 47.1 a | 6929.7 a | 44.2 a | 1370.7 d | 8.7 d |
| | | W3 | 7064.4 a | 45.9 a | 6796.6 a | 44.2 a | 1530.2 c | 9.9 c |
| | | W6 | 6391.8 b | 42.3 b | 6377.8 b | 42.2 b | 2329.4 b | 15.4 b |
| | | W9 | 6073.4 c | 41.5 b | 5875.2 c | 40.1 c | 2697.8 a | 18.4 a |
| | YN19 | W0 | 5603.7 a | 39.8 a | 7431.0 a | 52.8 b | 1032.9 d | 7.3 d |
| | | W3 | 5382.5 a | 38.6 a | 7185.8 a | 51.5 b | 1390.3 c | 10.0 c |
| | | W6 | 4508.1 b | 34.4 b | 6920.9 b | 52.8 b | 1686.3 b | 12.9 b |
| | | W9 | 4039.5 c | 31.4 c | 7001.5 a | 54.5 a | 1804.5 a | 14.0 a |
| 2016–2017 | YM18 | W0 | 8011.2 a | 47.5 a | 7042.7 a | 41.8 a | 1783.4 c | 10.6 c |
| | | W3 | 7807.3 a | 47.0 a | 6764.4 a | 40.7 a | 2016.5 b | 12.3 b |
| | | W6 | 7231.1 b | 46.6 a | 6451.9 b | 41.1 a | 2035.4 b | 12.8 b |
| | | W9 | 6887.9 c | 44.8 b | 6138.0 c | 39.9 a | 2326.7 a | 15.2 a |
| | YN19 | W0 | 7548.6 a | 46.7 a | 7045.5 a | 43.5 a | 1566.1 b | 9.7 d |
| | | W3 | 7440.7 a | 45.4 a | 7040.7 a | 43.0 a | 1888.1 a | 11.5 c |
| | | W6 | 6745.5 b | 43.7 b | 6778.8 b | 43.9 a | 1915.9 a | 12.4 b |
| | | W9 | 6038.8 c | 42.3 b | 6271.6 c | 43.9 a | 1952.6 a | 13.7 a |

Note: Different lowercase letters in the same column indicate significant differences in the duration of waterlogging in the same cultivar ($p < 0.05$).

### 3.6. Grain Yield and Its Components

Waterlogging treatments had no significant effect on the total number spikes in this study but other grain yield components such as kernel number, TGW, and final yield all were significantly affected by W9 treatment (Table 5). During 2015–2016, W6 had no significant effect on kernel numbers, but the wheat cultivars produced significantly fewer kernels under W6 than W0 during 2016–2017. Furthermore, YN19 and YM18 produced 8.9% and 8.2%, 12.5% and 10.0% fewer kernels per spike under W9 compared with their respective W0 plants during 2015–2016 and 2016–2017, respectively.

**Table 5.** Wheat yield and components under different treatments.

| Cultivar | Growing Season | Treatment | Number of Spikes (×10$^4$·hm$^{-2}$) | Kernels Per Spike | 1000-Grain Weight (g) | Yield (kg·hm$^{-2}$) |
|---|---|---|---|---|---|---|
| YM18 | 2015–2016 | W0 | 561.7 a | 38.0 a | 36.4 a | 7380.8 a |
| | | W3 | 553.6 a | 37.6 a | 36.2 a | 7064.4 a |
| | | W6 | 550.9 a | 36.4 a | 34.5 b | 6391.8 b |
| | | W9 | 540.0 a | 34.9 b | 32.0 c | 6073.4 c |
| | 2016–2017 | W0 | 587.4 a | 39.2 a | 35.9 a | 8011.3 a |
| | | W3 | 576.9 a | 38.3 a | 35.5 a | 7807.3 a |
| | | W6 | 575.7 a | 37.0 b | 33.5 b | 7231.1 b |
| | | W9 | 578.3 a | 35.3 b | 30.6 c | 6857.9 c |
| YN19 | 2015–2016 | W0 | 520.9 a | 31.6 a | 34.1 a | 5603.7 a |
| | | W3 | 505.4 a | 30.7 a | 33.8 a | 5382.5 a |
| | | W6 | 497.6 a | 30.5 a | 31.5 b | 4508.1 b |
| | | W9 | 495.2 a | 28.8 b | 29.8 c | 4039.5 c |
| | 2016–2017 | W0 | 594.0 a | 38.3 a | 33.0 a | 7548.6 a |
| | | W3 | 591.8 a | 37.8 a | 32.6 a | 7440.7 a |
| | | W6 | 580.5 a | 35.7 b | 30.4 b | 6745.6 b |
| | | W9 | 580.2 a | 33.5 c | 28.8 c | 6038.8 c |

Note: Different lowercase letters in the same column indicate significant differences in the duration of waterlogging in the same cultivar ($p < 0.05$).

Averaged across the two years, compared with W0 treatment, TGW of YM18 and YN19 was reduced by 6.1% and 7.9%, respectively under W6 treatment, while this reduction was 13.5% and 12.8%, respectively, under W9 treatment. Meanwhile, cultivar YM18 and

YN19 experienced 16.0% and 23.4% reduction in grain yield, respectively in response to W9 treatment compared with their respective W0 plants.

## 4. Discussion

### 4.1. Effect of Waterlogging on Photosynthetic Characteristics

Our study suggested that post-flowering soil waterlogging (>6 days) accelerated leaf senescence, damaged chlorophyll (SPAD value) and consequently reduced assimilation rates in the tested wheat cultivars. We also found that this damage to chlorophyll was irreversible, as this chlorophyll loss increased during later developmental stages after waterlogging was terminated. For example, compared with W0, W9 plants experienced 11.9%, 20.8% and 31.4% (averaged across the tested cultivars) reduction in leaf SPAD at 7, 14 and 21 days after anthesis, respectively. On average, waterlogging more significantly affected leaf SPAD in YN19 than in YM18. For example, W9 caused 12.7%, 24.3% and 33.2% reduction of SPAD in YN19 and 11.1%, 17.4% and 29.7% reduction in YM18 at 7, 14 and 21 days (Figure 2). Moreover, YM18 had a relatively higher net photosynthetic rate of flag leaves and less reduction under all waterlogged treatments, showing less damage of photosynthesis than YN19. This could be potentially because reactive oxygen species started accumulating as the oxygen supplies from root decreased [31].

Poor oxygen supply from the waterlogged soils induces reactive oxygen species generation, which can impair chlorophyll synthesis [32]. Similarly, poor root respiration in waterlogged soils can build up intercellular $CO_2$ concentration in leaves, thereby inhibiting photoreaction and reducing the net photosynthetic rate [33]. Leaf chlorophyll is highly sensitive to soil waterlogging, even during early stages of development. For example, chlorophyll loss in wheat leaves has been recorded without any visual signs of chlorosis or necrosis in response to soil waterlogging as early as during three to four leaf growth stages [34]. However, the significantly high sensitivity of leaf chlorophyll in wheat has been recorded in wheat crops in response to post-flowering waterlogging [35].

Waterlogging inhibited gas exchange between the roots and atmosphere so that the oxygen concentration decreases rapidly, and then, it accelerated leaf senescence [36]. The reduction in leaf greenness coincided with a reduction in stomatal conductance [24], and stomatal closure could constrain internal $CO_2$ levels and limit carbon fixation, which would seem to coincide with a decrease in photosynthesis rate [35]. A relatively superior performance of YM18 under waterlogged soils could be associated with an optimized root development, i.e., it maintained root length density which supported the oxygen supplied and photosynthesis in the flag leaf of waterlogging-tolerant wheat cultivars [37].

### 4.2. Effect of Waterlogging on Grain Weight and Grain Filling Rate

The early stage of grain filling includes fertilization and a period of rapid cell division when all seed structures are formed and the dry weight increases slowly, given that grain did not require abundant assimilate supplies during this phase [38]. However, poor assimilate supplies from leaves significantly affected the later stages of grain growth, when developing grains require continuous carbohydrate supply. For example, research showed that during the early days of grain filling (9–14DAA), the difference in grain-filling rate was not significant between drought-stressed and control plants in two wheat cultivars, but a significant difference was observed during the later grain filling days [39]. Meanwhile, compared with optimum temperatures, high temperatures enhanced the grain-filling rates slightly before 21 DAA, but they declined during later filling [40].

In this study, post-flowering waterlogging reduced grain yield by inhibiting grain developmental processes. For example, waterlogging (W6 or W9) significantly variably affected the grain filling rate and duration of the two tested cultivars and thus genotype performance. Averaged across two years, W9 accelerated the grain filling rate (i.e., increased the maximum grain filling rate (Vm) by 40.7%) but shortened the grain filling stage duration (i.e., 24.1% in YM18 cultivar). In contrast, the maximum grain filling rate (Vm) and mean grain filling rate (Va) were decreased by 18.4% and 18.6% in YN19 (Table 3). This indicated

that poor assimilates supplies, particularly later during grain development reduced the duration of the slow increase in grain filling rate in YM18 and final grain size. This was also supported by a significant reduction in TGW later during the development, i.e., 21 and 28 days after anthesis (Figure 4). Averaged across two years, under W6 and W9, TGW at maturity was reduced by 6.1% and 13.5%, respectively, in YM18 and 7.9% and 12.8%, respectively in YN19 (Table 5). This indicates that YM18 partially compensated the reduction of T and T3 by increasing Vm and T1, while YN19 reduced TGW due to the reduction of Vm and Va.

### 4.3. Effect of Waterlogging on Dry Matter Distribution

Post-anthesis waterlogging decreases carbon assimilation by damaging leaf pigmentation [3], inactivating starch synthesis enzymes in grains, and restricting nitrogen uptake in shoots [41]. This inhibits the dry matter accumulation and distribution in plants. Field studies suggested that a waterlogging event 7–16 days after anthesis can accelerate leaf senescence as well as the remobilization of water-soluble carbohydrates stored in culms [35]. In this study, compared with W0 plants, W6 and W9 of both cultivars significantly reduced dry matter accumulation in grain and stem + sheath and the proportion of dry matter accumulation in grain to total dry matter accumulation, while it increased dry matter accumulation in spike axis + glume and spike axis + glume in proportion to the total. However, compared with YN19, YM18 exhibited higher grain yield under the same waterlogging level in both growing seasons, suggesting a relatively higher level of waterlogging tolerance (Table 4). The reason might be that waterlogging after anthesis inhibited dry matter accumulation and the transport of dry matter into grains [42].

### 4.4. Effect of Waterlogging on Yield and Components

In this study, waterlogging had no significant effect on spike numbers but W6 and W9 significantly reduced TGW, although grain numbers were reduced only in response to W9 treatment. This suggests a relatively high sensitivity of developing grains to waterlogging at anthesis. Given that wheat plants could sustain short-term (3 days) waterlogging at anthesis, thus, grain numbers were not affected by 3–6 days of waterlogging. However, sustained waterlogging could induce ethylene accumulation [15] and thus terminated grains [43].

Grain yield was further declined under W9, which reduced the TGW and kernels per spike of YM18 (13.5%, 9.1%, respectively compared with W0) and YN19 (12.8%, 11.0%, respectively compared with W0). This grain yield reduction was the result of accelerated leaf senescence and inhibited photosynthesis in response to waterlogging. The plants did not show any growth recovery after the termination of soil waterlogging, indicating that the assimilation process was irreversibly damaged. Wheat crops are highly sensitive to soil waterlogging during anthesis. For example, 7 days of waterlogging at the booting, anthesis, and 15 days post-anthesis, reduced wheat grain yield by 18.4%, 41.8%, and 5.7%, respectively [44]. The grain yield reduction in response to waterlogging during this phase is mainly associated with reduced grain weight and grain set [45].

### 5. Conclusions

In this study, reduced TGW and grain yield in response to 6- and 9-days of waterlogging at anthesis was associated with impaired leaf greenness and inhibited the net photosynthetic rate of flag leaves at the late grain filling stage. This restricted assimilates supply and reduced dry matter accumulation in grain and stem + sheath and the proportion of dry matter accumulation in grain to total dry matter accumulation, resulting in significant grain yield loss at maturity. This study found that waterlogging variably affected the grain developmental process of two tested wheat cultivars. Furthermore, our study suggested that YM18 partially compensated for a waterlogging-induced reduction in T and T3 by sustaining assimilates supplies and Vm.

## 6. Key Findings

From the present study, it may be inferred that the grain yield reduction of both cultivars was the result of accelerated leaf senescence and inhibited photosynthesis in response to waterlogging. The plants did not show any growth recovery after the termination of soil waterlogging, indicating that the assimilation process was irreversibly damaged.

**Author Contributions:** Conceptualization, Z.H., X.H. and S.M.; formal analysis, P.G.; investigation, Y.W. and Y.S.; writing—original draft preparation, S.M. and P.G.; writing—review and editing, N.U., W.Z. and Y.F. All authors have read and agreed to the published version of the manuscript.

**Funding:** This research was supported by the National Natural Science Foundation of China (31801287), the grants from the National Key Research and Development Program of China (2017YFD0301306 and 2017YFD0301305), the National College Students' innovation and entrepreneurship training program of China (202110364044), and the Project of China Scholarship Council (201908775002).

**Conflicts of Interest:** The authors declare no conflict of interest.

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
