# Peer review of "Carbohydrate Assimilation and Translocation Regulate Grain Yield Formation in Wheat Crops (Triticum aestivum L.) under Post-Flowering Waterlogging"

_agronomy, doi:10.3390/agronomy11112209_

Round 1
Reviewer 1 Report
General comments
1. This is an important study for the journal agronomy.
Major comments
2. Figures 2 and 3. SPAD has a connection to photosynthetic rate. But it isn't perfect. I think most readers can trust this paper.
3. Figure 1. To write the wheat growth periods into that will get even better.
4. Table 5. Remove the mean values will be consistent like any other results. There is no point in writing. 5. Conclusions. I think the sentences would be generalized sophisticatedly.
Reviewer 2 Report
Paper is very interesting, I really liked the findings and the regression analysis of the grain filling rate. Al Please address these several points below and I will recommend the paper for publication.
L26: Pn not defined
L104-105: define normally
L126: was K defined post-harvest?
L127: Parameters A and B should be better defined
L182: i suppose these are the p-values
Table2: two-way ANOVA is described, yet the results show three-way analysis
Grouting just does not sound appropriate for this purpose. Why not filling?
Table 3 is very interesting; however it should be interpreted with regard to grain number per spike. In 2015-16 season, YM18 seems to favor W9 treatment, while in next year it does less. Ok, that is clear from table 5…
L394: Reduction?
Two paragraphs of conclusions regarding results are confusing. Iti s not clearly written is it reduction or increase.
